# Study of blood glucose and insulin infusion rate in real-time in diabetic rats using an artificial pancreas system

Omer Batuhan Kirilmaz[1◉], Akshay Radhakrishna Salegaonkar[2◉], Justin Shiau[2],
Guney Uzun[1], Hoo Sang Ko[1], H. Felix Lee[1], Sarah Park[3], Guim Kwon[2]*

1 School of Engineering, Southern Illinois University Edwardsville, Edwardsville, Illinois, United States of
America, 2 School of Pharmacy, Southern Illinois University Edwardsville, Edwardsville, Illinois, United
States of America, 3 Research and Instructional Services, Duke University, Durham, North Carolina, United
States of America

◉ These authors contributed equally to this work.
* gkwon@siue.edu

Molekulare Medizin Berlin Buch, GERMANY

**Data Availability Statement:** All relevant data are
within the paper.

**Funding:** GK (SIUE School of Pharmacy Research
Grants), HSK (SIUE internal awards).

## Abstract

Artificial pancreas system (APS) is an emerging new treatment for type 1 diabetes mellitus.
The aim of this study was to develop a rat APS as a research tool and demonstrate its appli-
cation. We established a rat APS using Medtronic Minimed Pump 722, Medtronic Enlite sen-
sor, and the open artificial pancreas system as a controller. We tested different dilutions of
Humalog (100 units/ml) in saline ranged from 1:3 to 1:20 and determined that 1:7 dilution
works well for rats with ~500g bodyweight. Blood glucose levels (BGL) of diabetic rats fed
with chow diet (58% carbohydrate) whose BGL was managed by the closed-loop APS for
the total duration of 207h were in euglycemic range (70–180 mg/dl) for 94.5% of the time
with 2.1% and 3.4% for hyperglycemia (>180mg/dl) and hypoglycemia (<70 mg/dl), respec-
tively. Diabetic rats fed with Sucrose pellets (94.8% carbohydrate) for the experimental
duration of 175h were in euglycemic range for 61% of the time with 35% and 4% for hyper-
glycemia and hypoglycemia, respectively. Heathy rats fed with chow diet showed almost a
straight line of BGL ~ 95 mg/dl (average 94.8 mg/dl) during the entire experimental period
(281h), which was minimally altered by food intake. In the healthy rats, feeding sucrose pel-
lets caused greater range of BGL in high and low levels but still within euglycemic range
(99.9%). Next, to study how healthy and diabetic rats handle supra-physiological concentra-
tions of glucose, we intraperitoneally injected various amounts of 50% dextrose (2, 3, 4g/kg)
and monitored BGL. Duration of hyperglycemia after injection of 50% dextrose at all three
different concentrations was significantly greater for healthy rats than diabetic rats, suggest-
ing that insulin infusion by APS was superior in reducing BGL as compared to natural insulin
released from pancreatic β-cells. *Ex vivo* studies showed that islets isolated from diabetic
rats were almost completely devoid of pancreatic β-cells but with intact α-cells as expected.
Lipid droplet deposition in the liver of diabetic rats was significantly lower with higher levels
of triacylglyceride in the blood as compared to those of healthy rats, suggesting lipid metab-
olism was altered in diabetic rats. However, glycogen storage in the liver determined by Peri-
odic acid-Schiff staining was not altered in diabetic rats as compared to healthy rats. A rat

Competing interests: The authors have declared that no competing interests exist.

APS may be used as a powerful tool not only to study alterations of glucose and insulin homeostasis in real-time caused by diet, exercise, hormones, or antidiabetic agents, but also to test mathematical and engineering models of blood glucose prediction or new algorithms for closed-loop APS.

## Introduction

The artificial pancreas system (APS) is an innovative new treatment for type 1 diabetes mellitus (T1DM), which reflects how advances in technology are incorporated in developing new treatment strategies for a human disease. An APS consists of medical devices designed to mimic the actions of insulin-secreting pancreatic β-cells which are selectively destroyed by autoimmune attack in T1DM patients. Pancreatic β-cells sense blood glucose levels (BGL) using an intricate metabolic signaling pathway [1] and release just the right amount of insulin into the portal vein and then to the circulation that maintain normal BGL in a healthy subject. In an APS a continuous glucose monitoring (CGM) sensor mimics the sensing mechanism of pancreatic β-cells. Currently available CGM sensors are far inferior to natural pancreatic β-cells in terms of accuracy and response rate. However, CGM sensors enable T1DM patients to control BGL much more effectively by reducing time in hypo- and hyperglycemia and increasing time in target range (70–180 mg/dl) [2]. Insulin pump mimics the action of insulin secretion by pancreatic β-cells. Recent advancement in technology enables patients to use sophisticated insulin pumps that regulate insulin delivery at any desired rate [3], closely mimicking the actions of pancreatic β-cells, although a lag in insulin action is still a challenge to overcome. An algorithm is a critical component of an APS, which determines appropriate insulin dosage needed to control BGL, accounting for the current glucose, rate of change of the glucose, and the amount of insulin that has already been delivered [4].

Despite some drawbacks and imperfections yet, an APS is an effective treatment option for T1DM, which reduces severe hypoglycemic episodes and long-term hyperglycemia-mediated diabetic complications [5]. An APS may also be used as a powerful research tool in a laboratory setting not only to study physiology of energy metabolism involving glucose and insulin homeostasis in real-time but also to provide a testing platform for proof of concept projects such as mathematical and engineering models of blood glucose prediction or new algorithms for closed-loop APS. Evaluation of the effectiveness of an anti-diabetic therapeutic agent often involves measurement of fasting or fed BGL [6], glucose or insulin tolerance test [7], glycosylated hemoglobin (HbA1c) [8], homeostatic assessment model (HOMA) of insulin resistance or insulin secretion [9], hyperinsulinemic-euglycemic clamp study [10], ex vivo studies of various organs/tissues, etc. Due to fluctuations of glucose and insulin levels throughout the day, interpretation of some of the data obtained using these methods has limitations in terms of accurate assessment of physiological alterations caused by the therapeutic agent. An APS allows studying blood glucose and insulin levels for 48–72 hours continuously with minimum invasiveness, reporting BGL and insulin infusion rate every 5 minutes, providing an overview of the physiological state in terms of glucose and insulin regulation in real-time. Biester et al. [11] reported that add-on therapy with a sodium glucose co-transporter 2 (SGLT2) inhibitor, dapagliflozin, under full closed-loop control, enhanced overall BGL control in adolescents and young adults with T1DM. Efficacy of dapagliflozin as an adjunct therapy for streptozotocin-induced diabetes may be evaluated using a rat APS. Evaluation of mathematical or computational modelling in physiology such as blood glucose prediction or glucose and insulin

regulation often uses in silico models. An APS will provide a testing platform that is more dynamic and closer to natural human physiology. In contrast to in silico models which often rely on the preset parameters of the virtual patients and make many assumptions, a rat APS test platform does not require making assumptions for modeling and simulation because it provides the input data on the fly. To this end, we have established a rat APS using Medtronic Minimed Pump 722, Medtronic Enlite sensor, and the open artificial pancreas system (Open-APS) as a controller. OpenAPS is a free open-source programming code that is developed by the community of the parents of type 1 diabetic children and type 1 diabetic patients. It is considered as the most advanced and rapidly evolving controller for closed-loop APS.

## Materials and methods

### Animals

Male Sprague Dawley rats (250-274g) were purchased from Harlan Sprague-Dawley Inc. (Indianapolis, IN) and were maintained in the animal facility at the School of Pharmacy at Southern Illinois University Edwardsville (SIUE) under controlled conditions (temperature 68–73˚F and 12 h light-dark cycle). The rats (3 per cage) were fed with Laboratory Rodent Diet 5001 (protein; 28.5%, carbohydrate; 58%, fat; 13.5% by weight) and water *ad libitum*. A total of 30 (18 diabetic and 10 healthy, 2 euthanized) rats were used for this study. The same rat was used 2–3 times in order to minimize inter-variability among rats and the number of rats used for the study for the majority of the experiments.

After completion of a study, rats were anesthetized using vaporized isoflurane (5% volume-volume) mixed with oxygen, a tail tip (1 mm) was cut, and blood (300–500 μl) was collected for assays. Rats were then injected intraperitoneally with 200 mg/kg pentobarbital and confirmed death by visualization of cessation of breathing and affirmation of no bodily movement. While the body is still warm, pancreas, liver, heart, and kidney were isolated for *ex vivo* studies and the remaining body was frozen in -20˚C prior to disposal. Out of 30 rats 2 rats were euthanized using vaporized isoflurane followed by injection of pentobarbital as described above to remove pain and suffering without collecting data because the rats showed distress after streptozotocin injection (closed eyes, no eating or drinking for 2 days, hunched in the corner of the cage). Severe hypoglycemia following streptozotocin injection might have caused the illness of the rats. After this incident, 10% sucrose water was given to rats for 1–2 days following streptozotocin injection until BGL >300 mg/dl was observed. With this treatment, no more animals were lost. All animal maintenance and treatment protocols complied with the Guide for Care and Use of Laboratory Animals as adopted by the National Institute of Health and approved by the SIUE Institutional Animal Care and Use Committee (IACUC).

### Chemicals

Laboratory Rodent Diet 5001 was purchased from EL Mel (St. Louis, MO) and Sucrose Dustless Precision Pellets were from Bio-Serve (Flemington, NJ). Insulin rabbit mAb was obtained from Cell Signaling (Danvers, MA). Stretozotocin and monoclonal mouse anti-glucagon antibody were obtained from Sigma (St. Louis, MO). The secondary antibodies, Alexa 647-conjugated donkey anti-rabbit antibodies and Alexa 488-conjugated donkey anti-mouse were obtained from Jackson Immuno Research Laboratories (West Grove, PA). Schiff reagent, periodic acid, total cholesterol, high density lipoprotein (HDL)-cholesterol, and triacylglyceride (TAG) liquid reagents and their respective standards were purchased from Fisher Scientific (Pittsburgh, PA) and alpha-amylase was from MilliporeSigma (Burlington, MA). Contour Next test strips were obtained from ADW Diabetes (Pompano Beach, FL). Rat hemoglobin A1c kit and control for the kit were obtained from Crystal Chem (Elk Grove Village, IL).

## Induction of streptozotocin-induced diabetes

After 6 hours of fasting, rats (~500 g) were anesthetized using vaporized isoflurane (5% volume to volume), followed by intraperitoneal injection with 60 mg/kg streptozotocin (~1 ml in 50 mM Na Citrate at pH of 4.5) using a 20G needle. Rats were considered diabetic when the BGL was greater than 300 mg/dl. All rats injected with streptozotocin developed full-blown diabetes in 1–3 days with BG greater than 500 mg/dl. Diabetic rats were injected daily with 2.5 units of Humalog in the morning and 2.5 units of Lantus in the evening to treat diabetes.

## Establishing and running a closed-loop APS experiment

Fig 1 shows an overview of a rat APS system. It consists of an insulin infusion pump (Medtronic Minimed Pump 722), a CGM (Medtronic Enlite sensor and transmitter), an Intel Edison computer-on-module loaded with OpenAPS, a laptop as a control server, an electronic food scale, and a restrainer to hold a rat. An Enlite sensor requires calibration 2h after insertion, followed by 6h initially, then every 12h. The sensor was calibrated a few times throughout the day following the manufacturer's instructions. For CGM sensor calibration, a lancing device (30G) was used to draw blood (1–2 μl) from the tail. OpenAPS (oref0 0.6.1) was installed on Edison/ Explorer following instructions in the OpenAPS Docs—Read the Docs (https://openaps. readthedocs.io/en/latest/). Nightscout (0.9.2) complemented with Heroku Cloud Application Platform was used to monitor BGL, insulin infusion rate, and OpenAPS looping using a laptop, office computer, or a smart phone device. A customized electronic scale with a linear actuator controlled by a micro-controller can measure food weight in any desired intervals. For

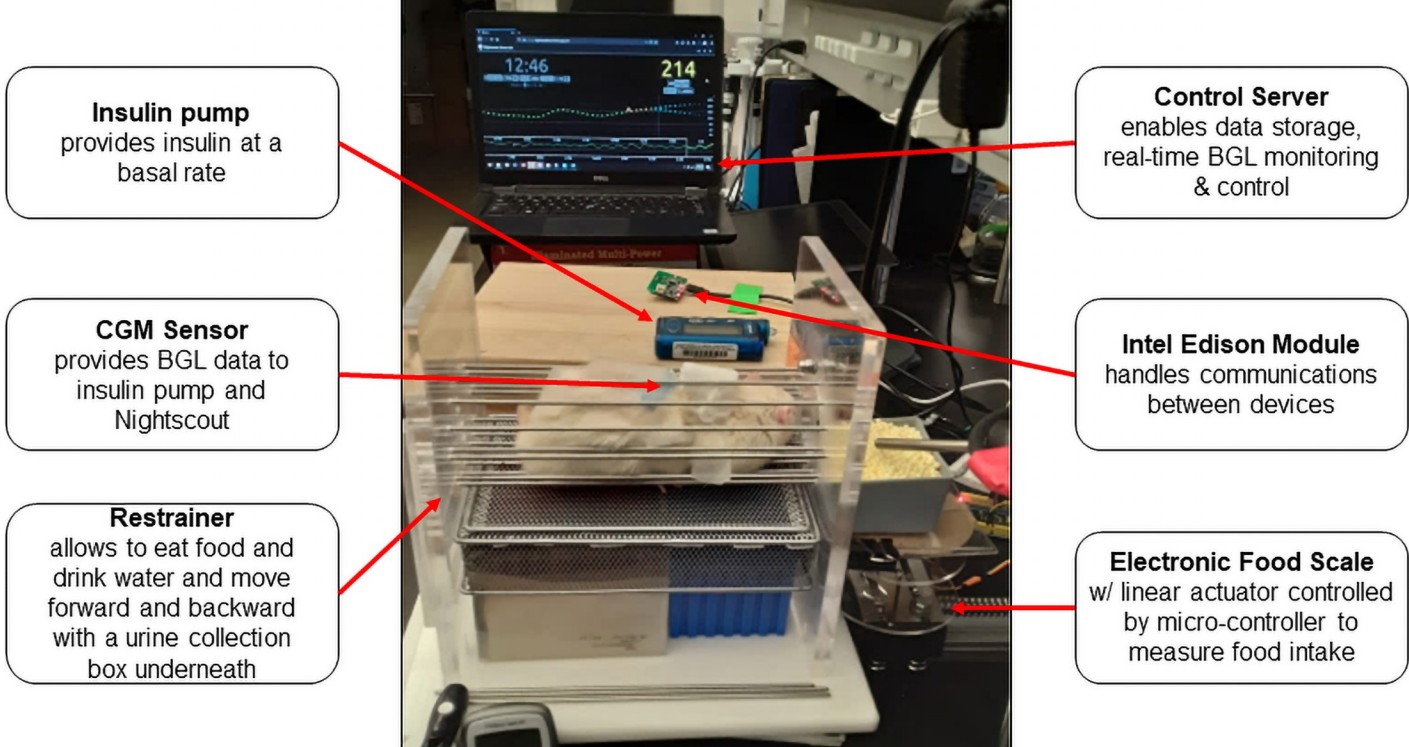

**Insulin pump**
provides insulin at a basal rate

**CGM Sensor**
provides BGL data to insulin pump and Nightscout

**Restrainer**
allows to eat food and drink water and move forward and backward with a urine collection box underneath

**Control Server**
enables data storage, real-time BGL monitoring & control

**Intel Edison Module**
handles communications between devices

**Electronic Food Scale**
w/ linear actuator controlled by micro-controller to measure food intake

**Fig 1. A photo image of a rat APS system.** The rat APS system consists of an insulin pump, a CGM, an Intel Edison loaded with OpenAPS, a laptop as a control server, an electronic food scale, and a restrainer to hold a rat.

this study, food weight was measured every 5 minutes to monitor food intake trend but the caloric intake was not incorporated into calculation of basal rate of insulin for this study.

For APS experiment, a rat was placed in a restrainer to secure CGM sensor and insulin needle. Each rat was allowed to acclimate to stay in the restrainer gradually (6h, 12h, overnight), taking a few days prior to experiment and the duration of an experiment was limited to maximum 3 days. We defined the humane endpoints including the inability to reach food or water for approximately 12 h, a 20% decrease in optimal body weight, a body condition score less than 3 on a 9-point scale, and significant pain that cannot be alleviated by analgesics. A rat placed in the restrainer was able to stretch head outside of the restrainer to eat food and drink water and move forward and backward. A rat also was able to sleep with head down on the bottom of the restrainer. Urination and bowel movement were normal. Rats, however, showed a mild sign of stress (slight porphyrin ring around eyes) by being confined in a small space despite the acclimation process. The porphyrin ring disappeared in a day after termination of an experiment. In order to minimize stress of being confined in a small space, an experiment was performed every other week for a rat for 2–3 days at a time. The frequency and the duration of experiment had not affected the health of a rat greatly based on the change of body-weight, eating, drinking, and grooming behaviors. Some experiments were cut short not due to concerns of animal welfare but technical issues such as sensor error, wifi connection, etc.

On the day of experiment, a diabetic rat was anesthetized using vaporized isoflurane, shaved an area (an inch diameter) behind front left paw for Medtronic Enlite sensor attachment site, and harnessed with Instech's adjustable belly bands with a vented dome. A small hole using a 18G needle was made on the dermis and a Medtronic Enlite Sensor was inserted and secured on to the rubber bands. A transmitter was attached to the sensor and let it stand against the vented dome to maintain the straight configuration of the sensor. A butterfly needle attached to the insulin pump line was inserted subcutaneously and secured using 3M Transpore tape. A rat attached with the sensor and insulin pump line then was placed in a Bowman Style Rodent Restrainer (Braintree Scientific Inc.). An Intel Edison loaded with the open source software (oref0 0.6.1) was used to communicate with CGM/pump and calculate appropriate basal rate of insulin based on glucose levels transmitted by the CGM. Cron, a time-based scheduling application in a Linux operating system was used to interact with the CGM and the pump to read glucose levels in real-time and enact the basal rate of insulin injection accordingly. To remotely monitor blood glucose levels and basal rates, Nightscout (0.9.2) complemented with Heroku Cloud Application Platform was used in conjunction with a Mongo DB instance. For a healthy rat, the same procedure described above was used except that the reservoir in the pump was filled with water instead of insulin and the insulin needle was placed in a microfuge tube, not in the body.

## Analysis of BGL and insulin infusion

The default Nightscout reporting view does not have a method allowing users to analyze data in a specific time range. A computer programmer in our team modified the Nightscout reporting view interface, allowing us to analyze BGL, closed-loop hours, insulin infusion rate/total insulin used, etc. for a specific time range.

## Intraperitoneal injection of dextrose

While a rat was attached with a Medtronic Enlite sensor and insulin line from the pump, intraperitoneal injection of 50% dextrose (2, 3, or 4 g/kg) with injection duration of ~5 sec was made through the bottom hole in the Bowman Style Rodent Restrainer and BGL was

monitored by Nightscout and recorded in the MongoDB. The data were extracted and area under curve was calculated using Microsoft Excel.

## Hemoglobin A1c assay

Tail blood was collected in a microfuge tube containing 15 mg of ethylenediaminetetraacetic acid (EDTA) and stored at 4°C. Glycated hemoglobin A1c (HbA1c) was determined using a rat HbA1c kit (Crystal Chem), following the manufacturer's instructions.

## Determination of plasma TAG, total-cholesterol, and HDL levels

After the studies that compared healthy vs. diabetic rats, rats were fasted for 12h and blood was collected into an EDTA-containing microfuge tube from the tail of a rat. Plasma was isolated by collecting supernatant after spinning blood for 20 min at 6000 rpm in a refrigerated microfuge centrifuge and stored at -20°C until use. Standard curve samples and triplicate of unknown samples (2 μl) were mixed with an appropriate liquid reagent (200 μl) in a 96 well plate. The samples were incubated at 37°C for 5 min, followed by absorbance reading at 500 nm using a Multiskan plate reader. For HDL cholesterol determination, the HDL fraction was first isolated by precipitating all beta-lipoproteins (LDL and VLDL) using 20% w/v polyethylene glycol, according to the manufacturer's instructions. The plasma concentrations of TAG, total-cholesterol, and HDL were determined using GraphPad Prism 6 statistics software.

## Frozen sectioning of whole pancreas and immunohistochemistry

Freshly isolated pancreases were placed in microfuge tubes, snap-frozen in liquid nitrogen, and stored at -70°C until use. Every 10th section of 10 μm thickness was collected and fixed in 4% paraformaldehyde and 1% Triton-X 100 in phosphate-buffered saline (PBS). Pancreas sections were then washed in PBS to remove the residual paraformaldehyde, blocked in 2% bovine serum albumin (BSA) in PBS, and treated with appropriate primary and secondary antibodies. 4′, 6-diamidino-2-phenylindole (DAPI) was used for nuclear staining. Fluorescent images were obtained using a 40X objective in an Olympus FluoView confocal microscope.

## Glycogen storage in the liver

Freshly isolated right lobe of the liver was snap-frozen in liquid nitrogen and stored at -70°C until use. Frozen section of 10 μm thickness was collected and fixed in 4% paraformaldehyde and 1% Triton-X 100 in PBS. Liver sections were washed in PBS to remove the residual paraformaldehyde and divided into two conditions with and without treatment with freshly prepared 0.5% alpha-amylase for 20 min. Liver sections were incubated with 0.5% periodic acid solution for 5 min, then stained with Schiff's reagent for 15 min. All steps were performed at room temperature, and liver sections were rinsed with distilled water after each step. Color images of liver sections were captured using a 20X objective in a Leica DMI inverted fluorescent microscope using ISCapture Software.

## Determination of lipid accumulation in the liver

Frozen liver sections of 10 μm thickness were cut using a Vibratome (St. Louis, MO) and every 10th section was transferred to a coverslip. Tissues were fixed and permeabilized in 4% paraformaldehyde and 1% Triton-X 100 in PBS. Tissues were washed in PBS to remove the residual glutaraldehyde, blocked in 2% BSA in PBS to reduce non-specific binding, and treated with Nile Red for lipid droplet and DAPI for nuclear staining. Fluorescent images were obtained using a 40X objective in an Olympus FluoView confocal microscope. Integrated fluorescent

intensity over number of nuclei for each liver section was determined using ImageJ image processing and analysis program.

## Statistical analysis

Statistical analysis was performed using GraphPad Prism 6 statistics software (t-tests and Mann-Whitney). We performed Mann-Whitney tests for comparing diabetic vs. healthy fed with chow or sugar pellets with respect to Normal % and Hypoglycemia % because these dependent variables were not normally distributed. Other than these 4 sets, the variables passed the normality test and thus we performed t-tests. Area under curve (AUC) was determined using Microsoft Excel program. Results are expressed as mean ± SEM. Significant differences are indicated by $^*$p<0.05, $^{**}$p < 0.01, and $^{***}$p<0.001, respectively.

## Results

### BGL and insulin infusion rate of a diabetic rat controlled by OpenAPS

Fig 2 shows a snapshot of Nightscout screen demonstrating BGL and insulin infusion rate of a diabetic rat controlled by OpenAPS. The rat ate food (chow diet) and drank water ad libitum. The green (80–180 mg/dl), yellow (56–80 mg/dl), and red dots (<55 mg/dl) represent optimal, borderline, and low BGL, respectively. The big red circles represent BG readings determined by Contour Next glucose meter, which were used to calibrate the sensor. These results demonstrate that BG readings by the glucose meter and CGM were close to each other, validating the accurate readings of BGL by CGM. The blue shaded boxes show basal insulin infusion rate

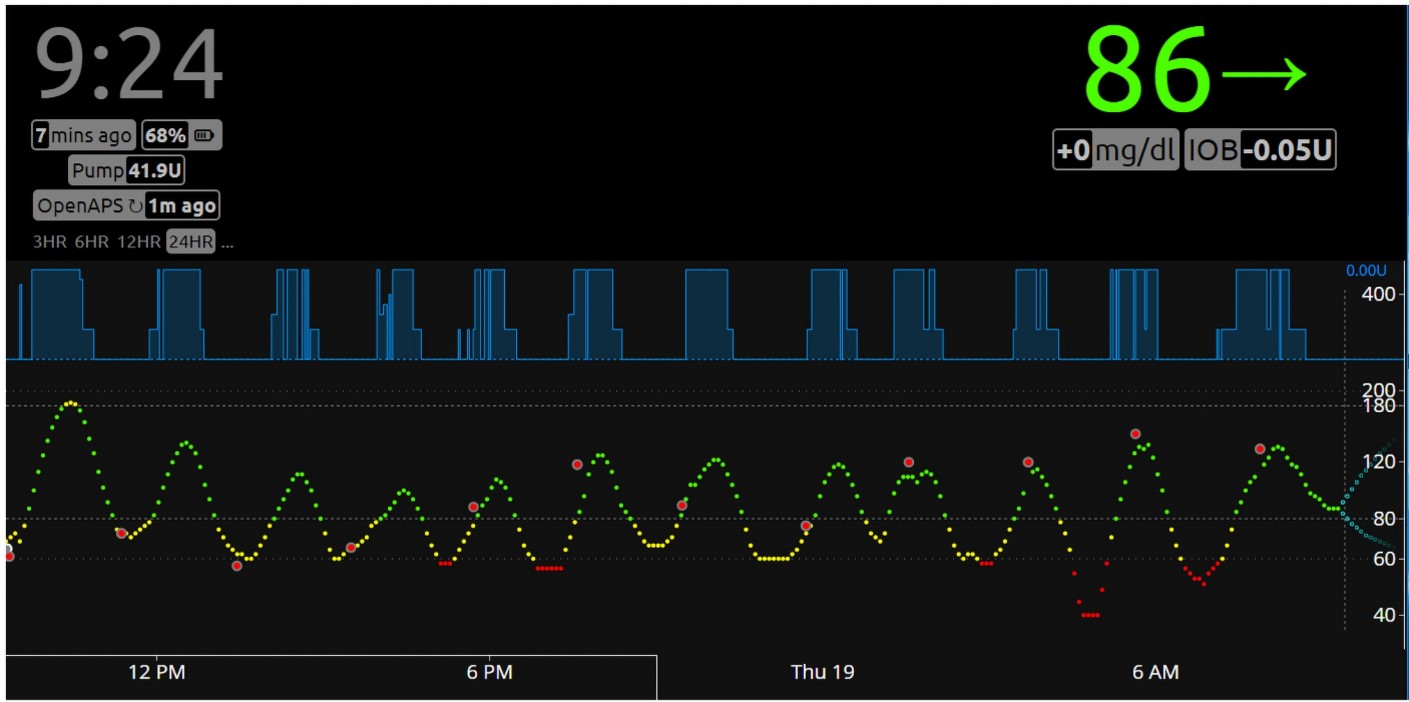

**Fig 2. BGL and insulin infusion rate of a diabetic rat controlled by OpenAPS.** A representative Nightscout image is shown with color-coded BGL including normal (green), moderate hypoglycemia (yellow), and severe hypoglycemia (red). The blue shaded boxes show basal insulin infusion rate controlled by OpenAPS. The big red circles represent BG readings determined by Contour Next glucose meter, which were used to calibrate the sensor. Other pertinent information such as current BG (86 in green), the trend of BG (horizontal arrow in green), the current time (9:24), sensor reading (7 min ago), and OpenAPS looping (1 min ago), etc. are also shown. The image shows a representative of 7 different experiments using two different diabetic rats.

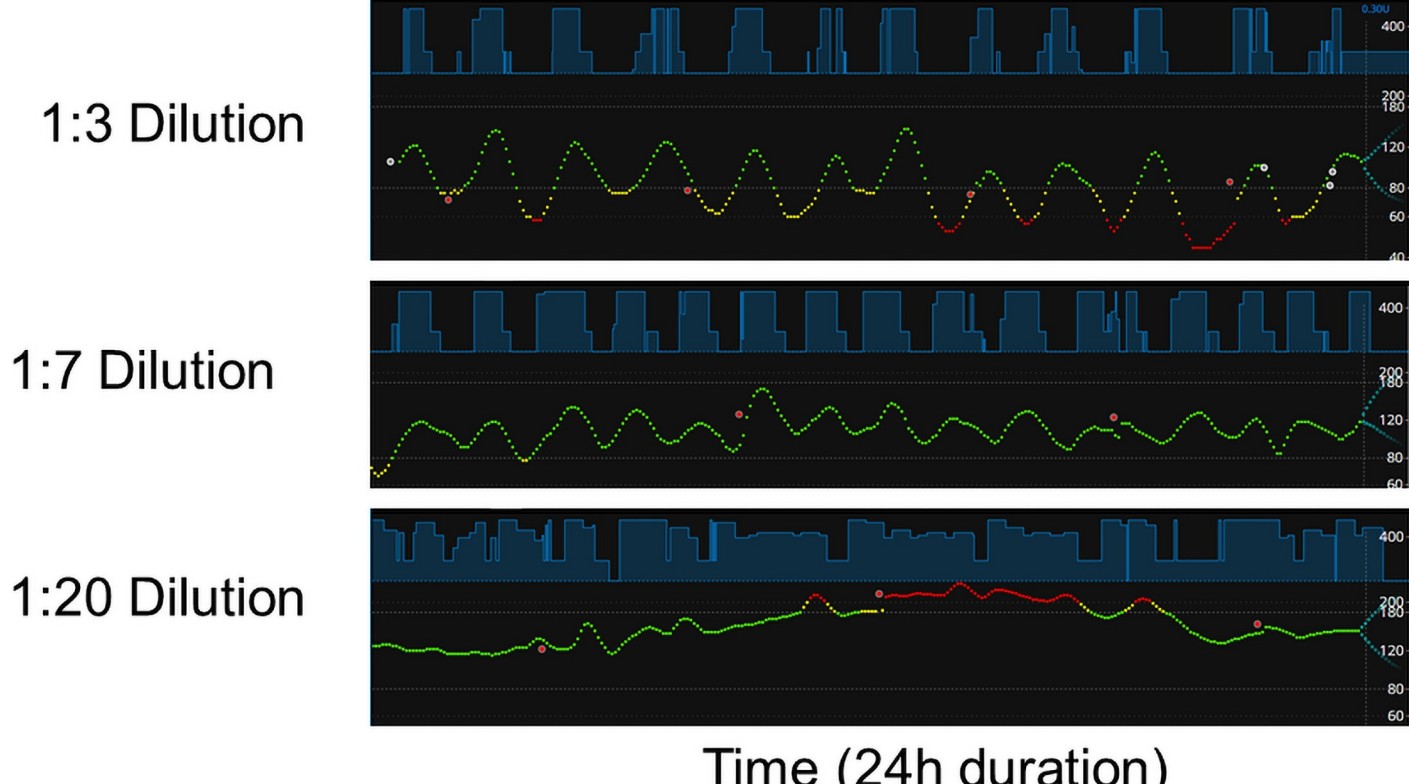

**Fig 3. Determination of an optimal concentration of Humalog for diabetic rats.** Representative Nightscout images of 1:3, 1:7, and 1:20 dilutions of Humalog are shown. OpenAPS controlled BGL was observed for the same diabetic rat using different dilutions of Humalog. Each individual experiment was run following the same procedure as described in the Methods section. A total of 20 independent experiments using 5 different diabetic rats and various dilutions of Humalog were performed.

controlled by OpenAPS. For this experiment, 1:3 dilution of Humalog (100 units/ml) in saline was used, which was too concentrated for the diabetic rat, resulting in hypoglycemia. It also shows the current BGL of 86 mg/dl in green at the upper right corner. The green horizontal arrow next to 86 indicates that the BGL is stable without up or down trend. Below are also shown no change in BGL (+0 mg/dl) and insulin on board (IOB) of -0.05U. In the upper left corner, the current time during the experiment (9:24 a.m.), the last CGM reading (7 mins ago), battery level (68%), insulin remaining in the reservoir (41.9U), the last OpenAPS looping (1m ago) and display window for 3, 6, 12, and 24HR are shown.

In order to optimize insulin concentration, we performed closed-loop OpenAPS experiments on the same diabetic rats using different dilutions of Humalog (100 units/ml) in saline. The rats were fed with chow diet and water ad libitum as the initial experiment shown in Fig 2. Fig 3 shows representative Nightscout images of 1:3, 1:7, and 1:20 dilutions of Humalog as labeled in the figure. 1:3 dilution of Humalog resulted in BGL overall in lower levels with some periods in a hypoglycemic (red dots, <55 mg/dl) range. 1:7 dilution of Humalog resulted in BGL within an optimal range (between 80 mg/dl and 180 mg/dl) for the majority of the experimental time period, while 1:20 dilution of Humalog resulted for an extensive period of time in a hyperglycemic range (red dots, >200 mg/dl) when the rat was eating actively during night time. With more diluted Humalog used, more frequent insulin infusions were observed shown by less gap between light blue boxes. For 1:20 dilution of Humalog, insulin was infused

continuously, indicating that OpenAPS was keep trying to bring BGL down. Based on the results of these experiments, we chose to use 1:7 dilution of Humalog for our subsequent experiments.

### Effects of different diets on BGL and insulin regulation in healthy and diabetic rats

To study the effects of different diets on BGL and insulin regulation, we performed closed-loop APS experiments using both healthy and diabetic rats. Fig 4 shows representative Nightscout images of diabetic rats fed with chow diet (upper panel) or sugar pellets (lower panel).

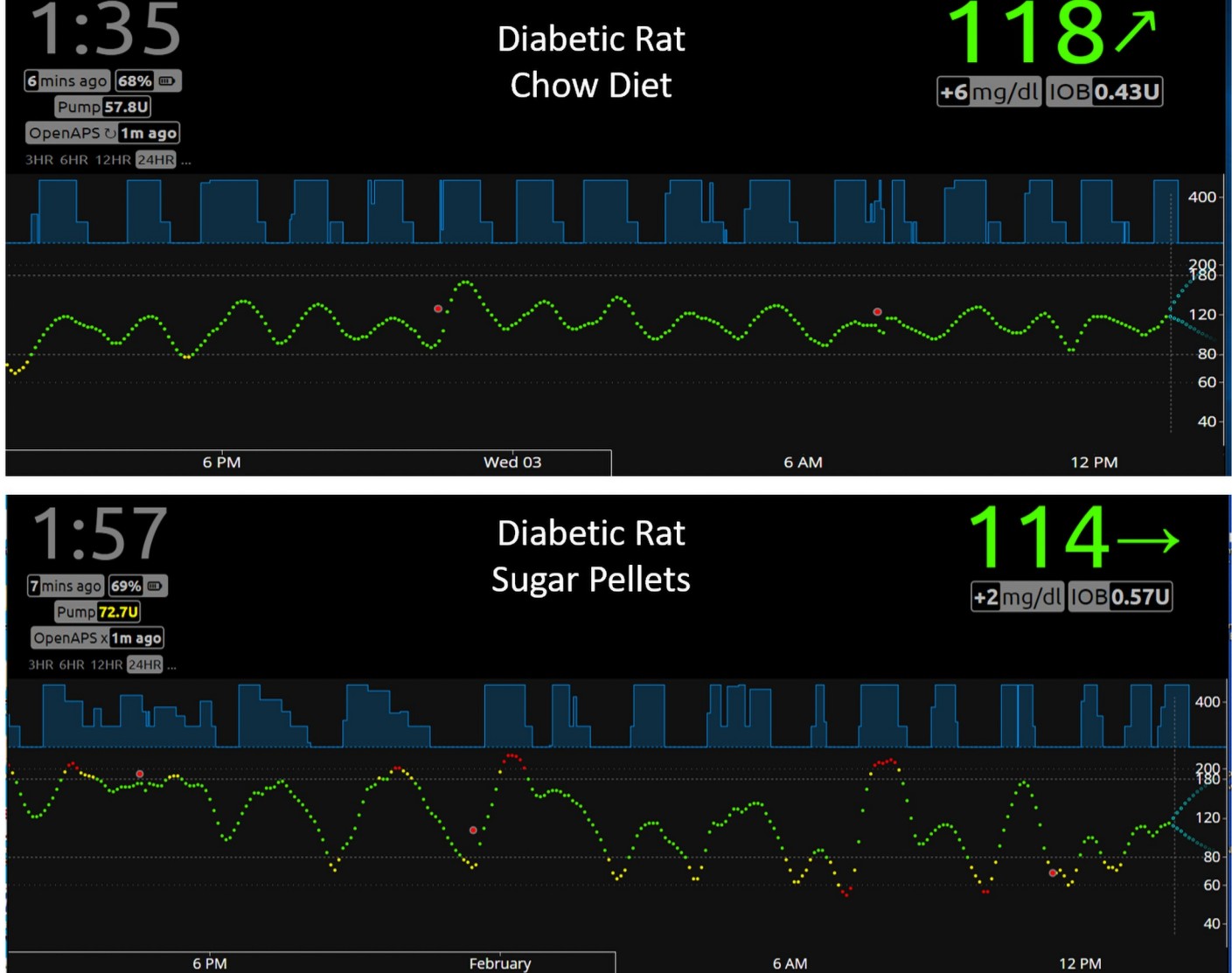

**Fig 4. Effects of different diets on BGL and insulin infusion in diabetic rats.** Representative Nightscout images of diabetic rats fed with chow diet (upper panel) or sugar pellets (lower panel) are shown. A total of 207 closed-loop hours (5 different rats with 10 independent experiments) using 1:7 dilution of Humalog was collected for diabetic rats fed with chow diet. Using the data analysis program described in the Methods section, the percent of time in hyper-, hypo- and normal glycemia as well as insulin infusion rate were calculated as shown in Table 1. A total of 175 closed-loop hours (3 different rats with 5 independent experiments) were collected for diabetic rats fed with sugar pellets (lower panel).

**Table 1. Summary of the data showing food type, closed-loop hour, and % BGL in hyperglycemia, normal, and hypoglycemia for healthy and diabetic rats.**

| Treatment Group | Food type | Closed-Loop (h) | Insulin Infusion (U/h) | Hyperglycemia (%, avg BG) | Normal (%, avg BG) | Hypoglycemia (%, avg BG) |
|---|---|---|---|---|---|---|
| Healthy Rats | Chow | 281 | 0.093 ± 0.012[#] | 0%, N/A | 99.9 ± 0.02%, 94.8 mg/dl | 0.67 ± 0.67%, 68 mg/dl |
| | Sugar Pellet | 146 | 0.114 ± 0.018[#] | 0%, N/A | 99.9 ± 0.08%, 112.3 mg/dl | 0.1 ± 0.08%, 61 mg/dl |
| Diabetic Rats | Chow | 207 | 0.138 ± 0.012* | 1.4 ± 0.83%***, 196.0 mg/dl | 95.1 ± 2.39%***, 122.6 mg/dl | 3.5 ± 2.34%, 65.4 mg/dl |
| | Sugar Pellet | 175 | 0.165 ± 0.014 | 31.1 ± 7.67%***, 243.2 mg/dl | 64.7 ± 7.55%**, 125.2 mg/dl | 4.3 ± 1.15%*, 62.9 mg/dl |

For healthy rats, insulin needle was placed in a microfuge tube, not in the body since they do not need insulin to control BG.

[#]denotes insulin infusion rate suggested by OpenAPS. Results are expressed as mean ± SEM.

* denotes statistical significance between healthy vs. diabetic rats for chow or sugar pellet.

As expected, rats fed with chow diet showed less fluctuations in BG levels than those fed with sucrose pellets. Table 1 shows that BGL levels of diabetic rats fed with chow diet for the duration of 207 closed-loop hours (5 different rats with 10 independent experiments) were in euglycemic range (70–180 mg/dl) for 95.1% of the time with 1.4% and 3.5% for hyperglycemia (>180mg/dl) and hypoglycemia (<70 mg/dl), respectively. Diabetic rats fed with Sucrose pellets for the duration of 175 closed-loop hours (3 different rats with 5 independent experiments) were in euglycemic range for 64.7% of the time with 31.1% and 4.3% for hyperglycemia and hypoglycemia, respectively. Average insulin infusion rate for diabetic rats fed with chow diet was 0.137 U/h, whereas that for diabetic rats fed with sugar pellets was 0.165 U/h, indicating that OpenAPS infused more insulin in response to high BGL caused by consuming sucrose pellets as anticipated.

Heathy rats fed with chow diet showed almost a straight line of BGL ~ 95 mg/dl (average 94.8 mg/dl) during the entire experimental period of 281h (4 different rats with 8 independent experiments), which was minimally altered by food intake (Fig 5). It is noteworthy how the patterns of OpenAPS-suggested insulin infusion (mock insulin infusion) based on BGL shown in light blue in Fig 5 upper and lower panels are quite different from those in diabetic rats (Fig 4). Suggested insulin infusion rates by OpenAPS were lower in healthy rats than diabetic rats (0.093 U/h and 0.137 U/h vs. 0.114 U/h and 0.165 U/h for chow and sucrose pellets, respectively) as expected. In addition, suggested insulin infusions by OpenAPS in healthy rats regardless of food type were almost continuous, while those in diabetic rats were intermittent. Feeding sucrose pellets to healthy rats caused greater range of BGL but still within euglycemic range (99.9%).

## BGL after intraperitoneal injection of 50% dextrose in diabetic and healthy rats

Fig 6A and 6C show representative Nightscout images for a healthy and a diabetic rat, respectively, injected with 4 g/kg 50% dextrose intraperitoneally. The red arrow shows the time of injection. BGL rose rapidly, reaching above 400 mg/dl with elevated BGL for the duration of ~4 h and ~ 3h for the healthy and the diabetic rats, respectively, as shown in Fig 6B and 6D. The upper limit of BGL measurement for Enlite sensor is 400mg/dl, which is the reason why the plateau of 400 mg/dl are shown. Fig 6E shows area under curve for BGL, indicating that healthy rats showed two-fold higher than diabetic rats. Fig 7A panels a and b show representative Nightscout images of BGL for healthy rats injected with 2 g/kg and 3 g/kg 50% dextrose, respectively, while panels c and d show those for diabetic rats. Fig 7B shows area under curve for each condition shown in Fig 7A, indicating that healthy rats showed significantly higher

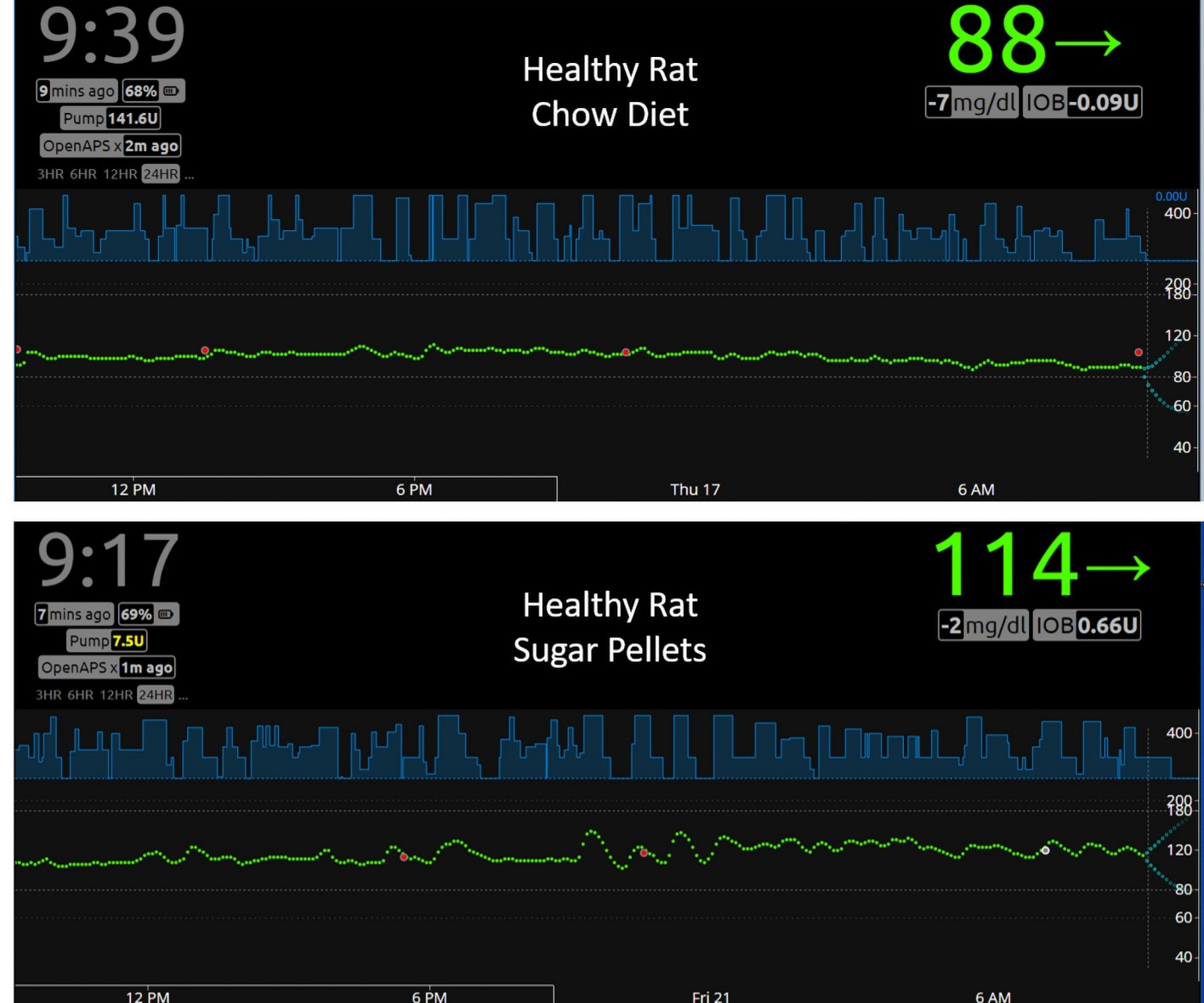

**Fig 5. Effects of different diets on BGL and suggested insulin infusion by OpenAPS in healthy rats.** Representative Nightscout images of healthy rats fed with chow diet (upper panel) or sugar pellets (lower panel) are shown. A total of 281 closed-loop hours (4 different rats with 8 independent experiments) was collected for healthy rats fed with chow diet. Using the data analysis program, the percent of time in hyper-, hypo- and normal glycemia as well as OpenAPS-suggested insulin infusion rate were calculated as shown in Table 1. A total of 146 closed-loop hours (3 different rats with 5 independent experiments) were collected for diabetic rats fed with sugar pellets (lower panel).

overall BGL compared to diabetic rats. Taken together, these results suggest that insulin infusion by APS was superior in reducing BGL as compared to natural insulin released from pancreatic β-cells as anticipated since there is no limitation of insulin secretion or reservoir.

## Assessment of physiological states of diabetic and healthy rats

To understand the physical states of diabetic rats in terms of pancreatic β-cell reserve and glycogen storage as an indicator of liver function, *ex vivo* studies were performed. Fig 8 shows

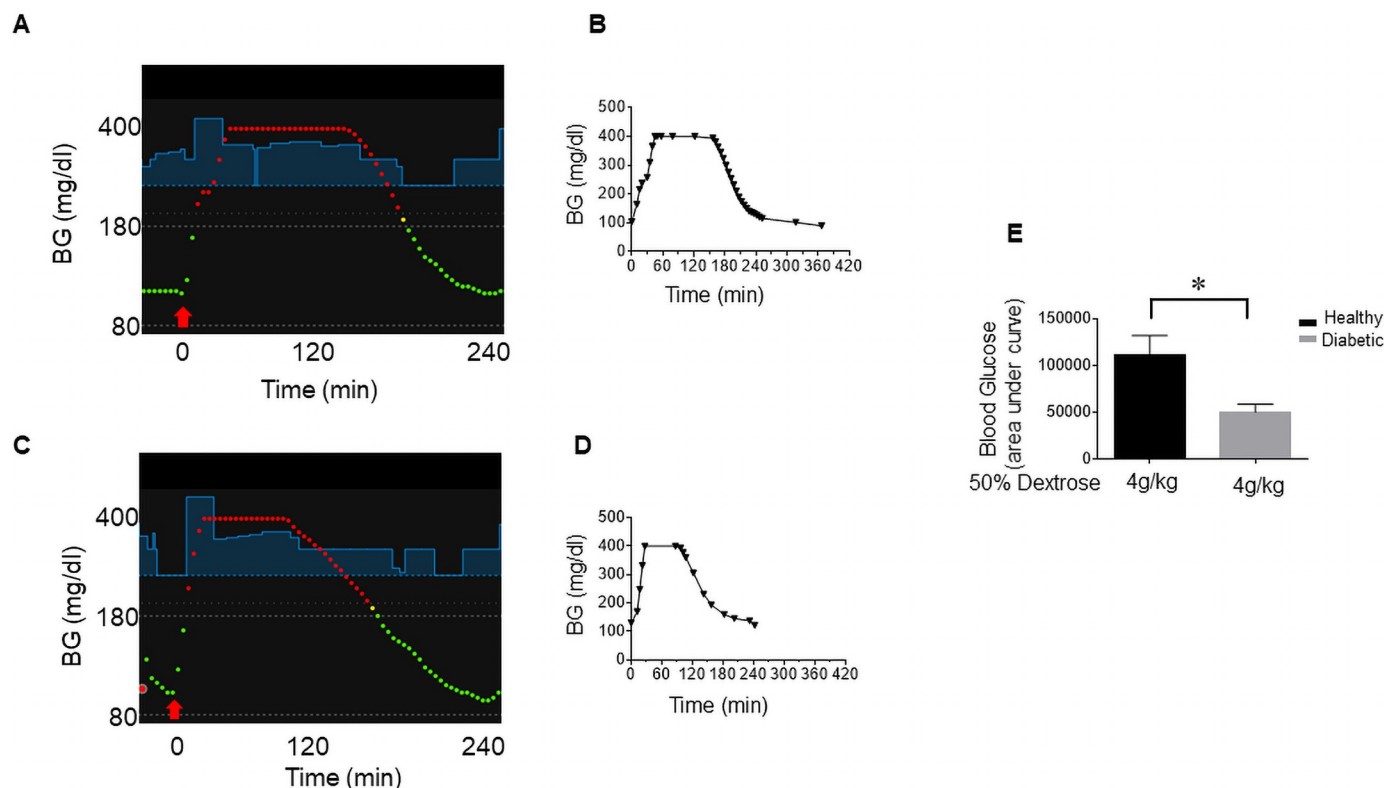

**Fig 6. BGL after intraperitoneal injection of 4 g/kg 50% dextrose in healthy and diabetic rats.** (A) and (C) Representative Nightscout images of BGL for healthy (n = 3) and diabetic (n = 3) rats, respectively, are shown. The red arrow shows the time of injection. Blue lines show suggested insulin infusion by OpenAPS. BGL above 200 mg/dl and below 180 mg/dl are shown in red and green dots, respectively. (B) and (D) BGL determined by Contour Next glucose meter (closed circles) and Enlite sensor (closed inverted triangles) for healthy and diabetic rats, respectively. (E) Area under curve for BGL determined by the glucose meter for healthy and diabetic rats. Values are presented as means ± SEM (n = 3).

frozen pancreas sections immunostained for insulin and glucagon to localize β- and α-cells. Fig 8 panels a-e demonstrate that islets embedded in the pancreases isolated from healthy rats showed localization of β-cells (red) in the center and α-cells (green) in the periphery. Fig 8 panels f-j show that islets in the pancreases isolated from diabetic rats were almost completely devoid of pancreatic β-cells but with intact α-cells. Moreover, the architecture of islet structure has been significantly altered with α-cells clustered together rather than lining around a circular shape as shown in Fig 8 panels a-e.

Fig 9A panels a and b show Periodic acid-Schiff (PAS) staining of frozen liver sections isolated from healthy and diabetic rats, respectively, to assess glycogen storage in the liver. Fig 9A panels c and d show the same conditions as panels a and b but the liver sections were stained with PAS after treatment with amylase which breaks the bonds in glycogen and removes it from the liver sections as a negative control condition. The liver sections treated with amylase (panels c and d) show much lighter magenta coloring due to removal of glycogen. Fig 9B shows integrated intensity of the staining, which shows higher values in the conditions treated with amylase due to more light penetration through relatively lighter magenta coloring. Taken together, these results show that liver function in terms of glycogen storage was not altered in diabetic rats as compared to healthy rats.

Fig 10A panels a and b show lipid accumulation in the liver of healthy and diabetic rats, respectively, determined by Nile Red staining. Fig 10B shows integrated fluorescent intensity over number of nuclei for each liver section using ImageJ program. Lipid accumulation in the

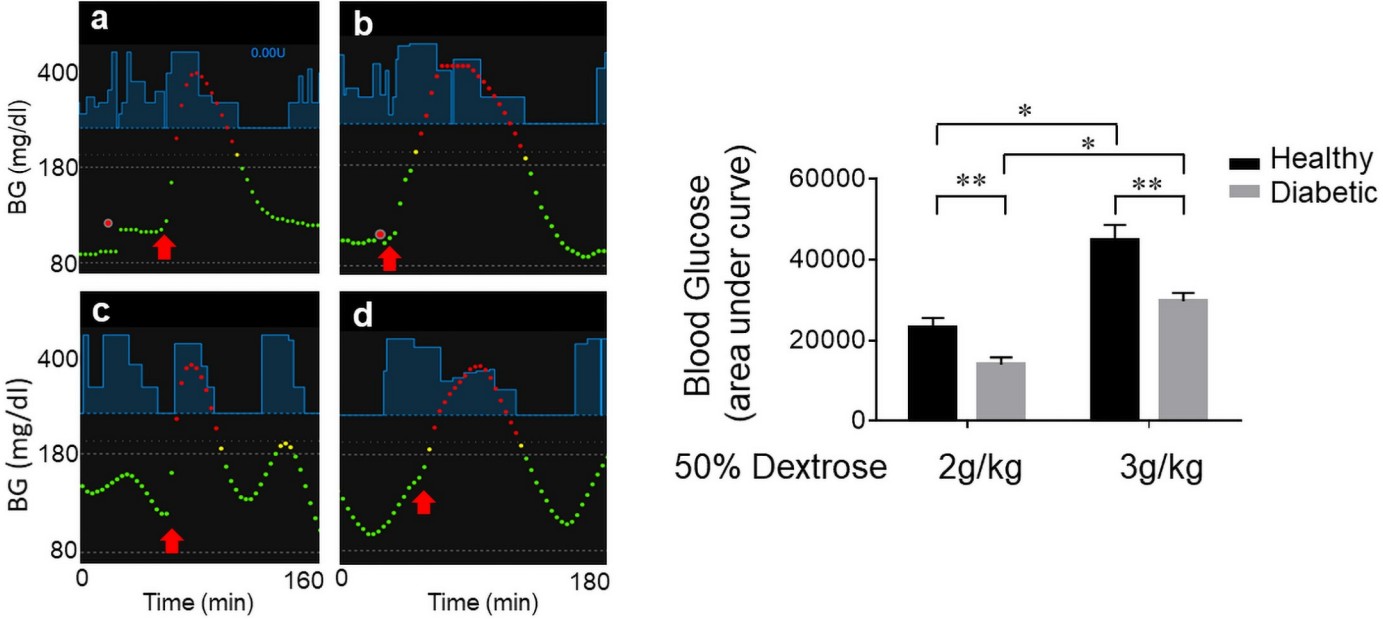

**Fig 7.** (A) BGL after intraperitoneal injection of 2 g/kg and 3 g/kg 50% dextrose in healthy and diabetic rats. Panels a and b show representative Nightscout images for BGL for healthy rats (n = 3) injected with 2 g/kg and 3 g/kg 50% dextrose, respectively, and Panels c and d show those for diabetic rats (n = 3). (B) Area under curve for BGL determined by Enlite sensor for each condition as indicated on the figure. Values are presented as means ± SEM (n = 3).

liver of the diabetic rats was significantly lower than that in the healthy rats. TAG and HDL in the blood, on the other hand, were significantly higher in diabetic rats than healthy rats (Table 2). We have also measured HbA1c levels of healthy and diabetic rats as shown in Table 3. The diabetic rats showed significantly higher HbA1c levels than healthy rats.

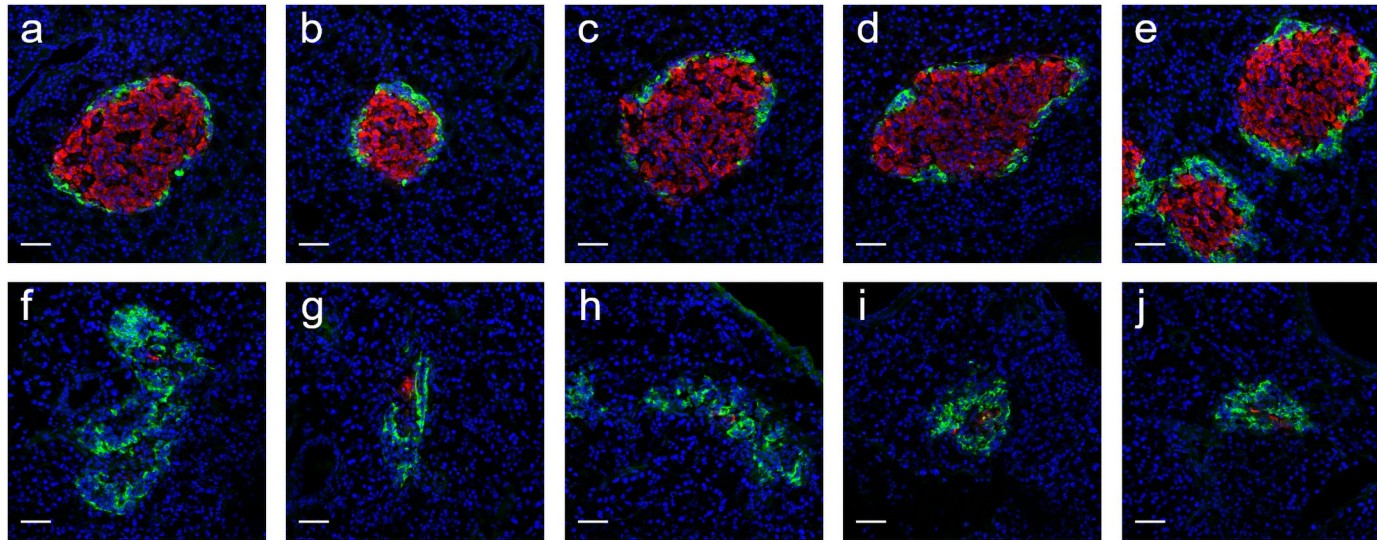

**Fig 8. Representative images of islets from healthy and diabetic rats.** Panels a-e (healthy) and f-j (diabetic) show frozen pancreas sections immunostained for insulin (red) and glucagon (green) to localize β- and α-cells, respectively. Pancreas sections (n = 3 for each condition) were processes for immunostaining using appropriate primary and secondary antibodies as stated in the Methods section. DAPI was used for nuclear staining (blue). Florescent images were obtained using a 40X objective in an Olympus FlowView confocal microscope.

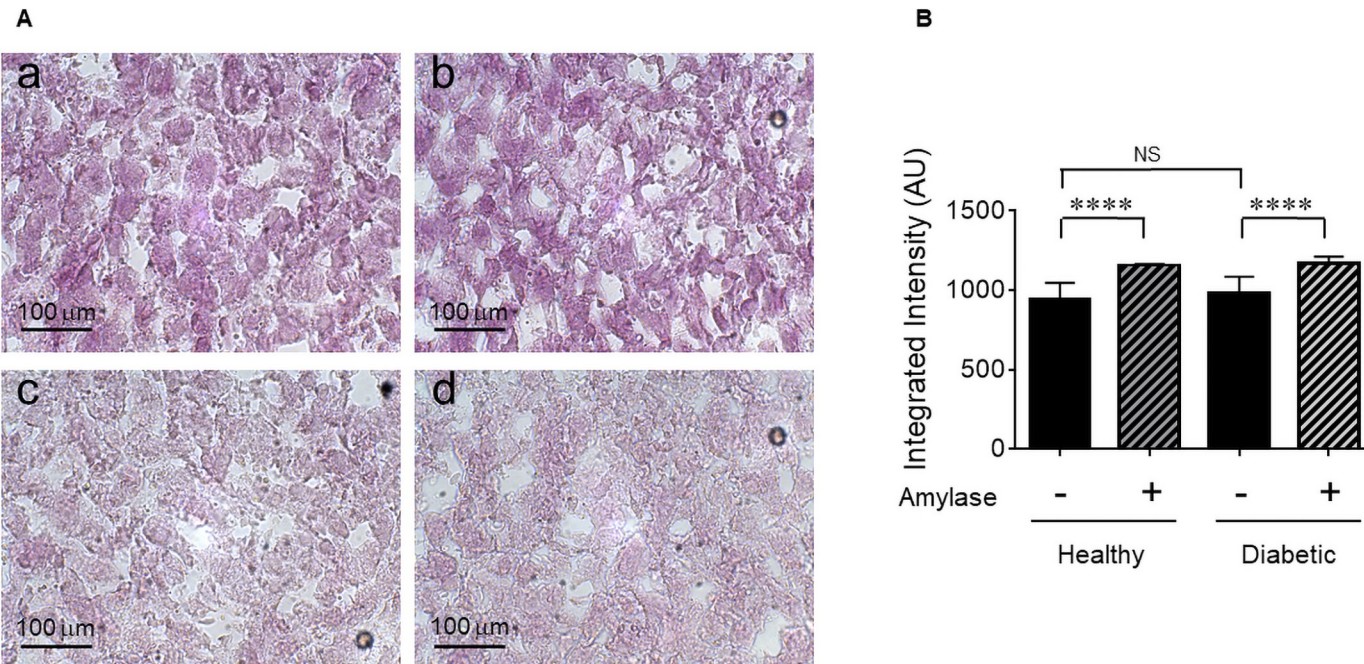

**Fig 9. Assessment of glycogen storage in the liver.** (A) Panels a and b show representative images of Periodic acid-Schiff (PAS) staining of frozen liver sections isolated from healthy and diabetic rats, respectively. Panels c (healthy) and d (diabetic) show the images of PAS staining after amylase treatment as a control. (B) Integrated intensity of the PAS staining with and without amylase. Data are the means ± SEM of ~26–30 liver sections from 3 rats per condition.

## Discussion

In this paper, we have established a rat APS as a research tool not only to study alterations of glucose and insulin homeostasis in real-time caused by diet, exercise, hormones or antidiabetic agents, but also to test mathematical and engineering models of blood glucose prediction or new algorithms for closed-loop APS. One of the major challenges of developing algorithm for BGL prediction or closed-loop APS is a limited selection of validation models that often include virtual patients in silico, retrospective patient data, or a very small number of patients in restrictive settings such as hospital or hotel. A rat APS provides a versatile and dynamic *in vivo* testing platform that allows to study the impact of APS algorithm in controlling BGL of a rat in real-time.

Due to significant differences in the body weight of rat vs human, we performed closed-loop APS experiments using different dilutions of Humalog (100 units/ml) to optimize the insulin stock solution for a rat APS. For a rat, the strength of stock solution made a significant difference in the pattern of insulin infusion as well as BGL fluctuation. As anticipated, the higher strength of stock solution (Fig 3, 1:3 vs 1:7 and 1:20 dilution), the longer duration of interval between insulin infusion and the higher frequency of hypoglycemia were observed. For a rat with the body weight of ~500g, 1:7 dilution of Humalog was appropriate. The 1:20 dilution of Humalog caused continuous insulin infusion with no cyclic pattern of BGL shown with 1:3 or 1:7 dilution of Humalog. Of note, the BGL pattern of a healthy rat showed hardly any fluctuation with continuous insulin infusion albeit mock insulin infusion determined by the OpenAPS algorithm (Fig 5, upper panel). These results support the strategy of frequent supermicrobolus injections of insulin as a part of BGL control algorithm [12].

The effects of different diets on glucose-insulin regulation in healthy and diabetic rats were studied using a rat APS. There are stark differences in terms of insulin infusion and BGL

**A**

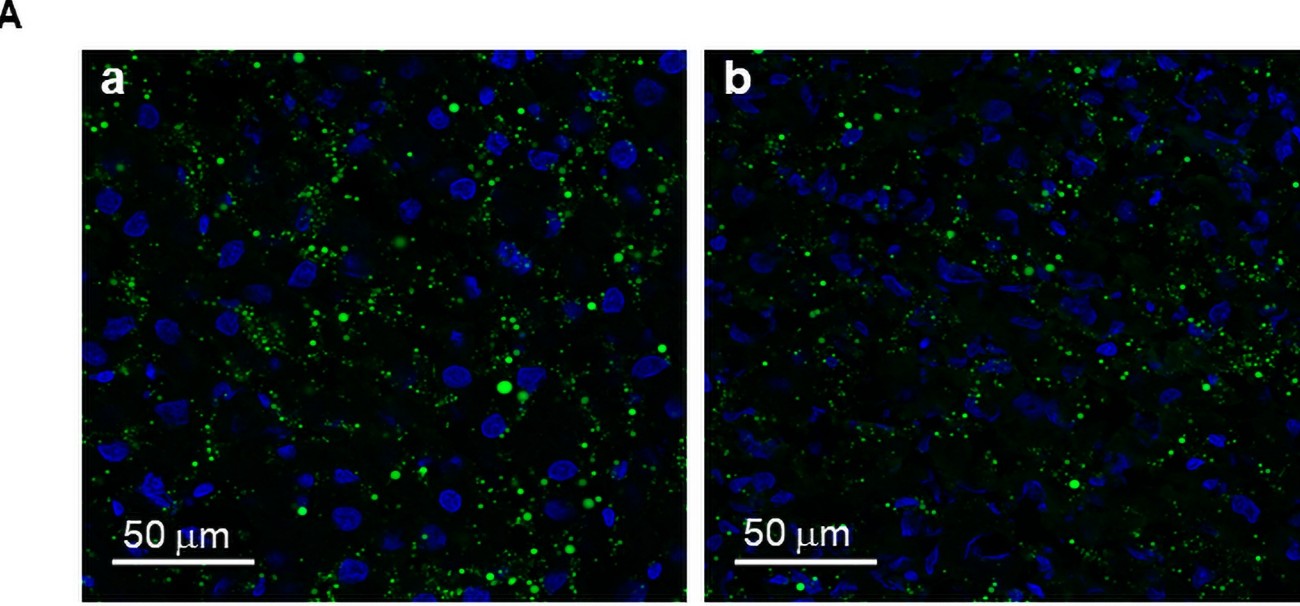

**B**

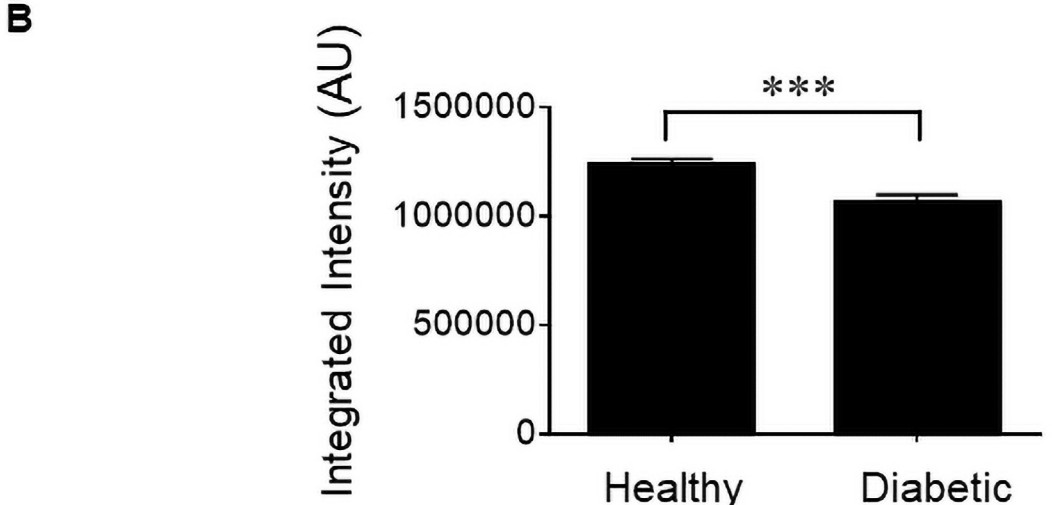

**Fig 10. Assessment of lipid accumulation in the liver.** (A) Panels a and b show representative images of Nile red staining of frozen liver sections isolated from healthy and diabetic rats, respectively. Fluorescent images were obtained using a 40X objective in an Olympus FluoView confocal microscope. (B) Integrated fluorescent intensity over number of nuclei for each liver section was determined using ImageJ image processing and analysis program. Data are the means ± SEM of ~26–30 liver sections from 3 rats per condition.

**Table 2. Plasma TAG, total cholesterol, and HDL levels of healthy and diabetic rats (n = 5).**

| Treatment Group | TAG (mg/dl) | Total Cholesterol (mg/dl) | HDL (mg/dl) |
|---|---|---|---|
| Healthy Rats | 75.7±6.5 | 63.1±3.7 | 34.3±4.8 |
| Diabetic Rats | 194.5±14.9**** | 83.8±8.7 | 103.6±16.3**** |

Colorimetric assays were performed to determine TAG, total cholesterol, and HDL in the plasma of the rats as stated in the Methods section.

**Table 3. Body weight and HbA1c levels of healthy and diabetic rats.**

| Treatment Group | Body Weight (g) | HbA1c (%) |
|---|---|---|
| Healthy Rats | 481±17 | 5.8±0.1 |
| Diabetic Rats | 480±30 | 7.2±0.3*** |

Body weight of diabetic and healthy rats were measured (n = 5). Tail blood was collected in a tube containing EDTA and stored at 4˚C. Glycated hemoglobin A1c (HbA1c) was determined using a rat HbA1c kit (Crystal Chem), following the manufacturer's instructions.

patterns between healthy and diabetic rats (Figs 4 and 5). Regardless of food type, BGL was completely controlled within the target range (70–180 mg/dl) in a healthy rat (Fig 5), showing how well intact pancreas controls BGL with a very small margin of error. Interpretation of the data, however, should be made cautiously. For the diabetic rats, the BGL patterns reflect the action of insulin infused externally from the pump in addition to other physiological factors. For the healthy rats, the actual amount or frequency of insulin released from the intact pancreas is unknown and the mock insulin infusion data determined by OpenAPS do not reflect the action of insulin unlike in the case of diabetic rats.

To study how well the OpenAPS algorithm handles supra-physiological concentrations of glucose compared to pancreatic β-cells, we injected various amounts of 50% dextrose intraperitoneally to diabetic and healthy rats (Figs 6 and 7). For intraperitoneal glucose tolerance test, 2 g/kg of glucose is commonly used [13]. To observe significant changes in the glucose-insulin regulation, up to 4 g/kg of 50% dextrose was injected. Interestingly, insulin infusion to diabetic rats by APS was superior in reducing the duration of hyperglycemia as compared to insulin released from native pancreatic β-cells in healthy rats. The study, moreover, provided insights how healthy rats respond to challenges of different amounts of glucose. Both the duration and the extent of hyperglycemia (area under curve) showed non-linear exponential relationship to different glucose amounts (i.e. 2, 3, 4 g/kg and 86 min, 161 min, 325 min, respectively, for the duration of hyperglycemia and 23,243 mg/dl min, 44,946 mg/dl min, 111,288 mg/dl min, respectively, for area under curve for BGL). Despite decades of research, it is still not clear how insulin release from millions of islets physically set apart in the pancreas of human or animal is coordinated to maintain BGL for a very small margin of error (~2–3 mM) on the hypoglycemic side of the blood glucose set point [14]. Upon glucose challenge, insulin secretion from the pancreas is reported to be a very small portion (<1%) of the total insulin stored in the pancreas [15]. Recent studies by Zhu et al. [16] provided evidence that only a partial fraction of islets in mice release insulin to the point of complete depletion of their storage in response to oral or intravenous glucose administration, suggesting that the majority of the islets are dormant. It is not known by which underlying mechanisms regulate insulin release from certain islets but not others. It is understandable that the duration and the extent of hyperglycemia are affected by many factors including insulin release from the pancreas, the rate of glucose uptake, the target cell mass, etc. The significant differences between the healthy and the diabetic rats despite their comparable body weights, however, suggest that there is a temporal or physical barrier in insulin secretion in the healthy rats. It is of great interest to understand insulin and glucose regulation in normal or disease states, which may help discovering therapeutic agents that enhance insulin release from the dormant islets in the setting of type 2 diabetes mellitus. The rat APS will be a valuable tool in studying alterations of glucose and insulin regulation in both type 1 and type 2 diabetic rats as compared to healthy rats.

Normal healthy rodent islets contain β-cells in the center and α-cells in the periphery of the islet [17,18]. Streptozotocin is taken up by pancreatic β-cells, causing its destruction [19]. The

glucose moiety in its structure directs its selective uptake through glucose transporter 2 (GLUT2) expressed abundantly on β-cells [20]. The cytotoxic effects of streptozotocin is caused by DNA strand breaks by the highly reactive methylnitrosourea moiety of the drug, followed by over-activation of poly-ADP ribose polymerase, $NAD^+$ depletion, reduction in cellular ATP, loss of membrane integrity, and necrotic β-cell death [21,22]. Dead β-cells are phagocytosed by macrophages, leaving α-cells behind. Fig 8 panels f-j show drastic alterations in the morphology of islets after destruction of β-cells, leaving α-cells clustered together. The complete or near-complete destruction of β-cells reflect the full-blown diabetes in rats injected with streptozotocin. The extent by which the remaining β-cells release insulin is unknown, but considered negligible based on the absence of β-cells in the diabetic rats shown in Fig 8.

Cytotoxic effects of streptozotocin on liver and kidney can occur because the cells in these organs also express GLUT2 despite their much lower levels than pancreatic β-cells [23]. In addition, hyperglycemia caused by diabetic condition can also cause damage to liver and kidney. Fig 9A and 9B, however, showed that daily injection of insulin to diabetic rats minimized the liver damage in terms of glycogen storage that allows glucose release from the liver between meals and during sleep. Liver plays a central role in lipid and glucose hemostasis [24]. In healthy subjects, the liver processes large quantities of lipids, but stores only small amounts in the form of TAG, with steady state TAG contents of less than 5% [25]. Excess TAG accumulation in the liver occurs in the setting of obesity, insulin resistance, and type 2 diabetes [26,27]. In streptozotocin-induced diabetic rats, a consistent increase in hepatic TAG content was observed only in the ketotic diabetic rats with no insulin treatment [28]. Our study shows that daily insulin treatment of the diabetic rats prevented hepatic steatosis. However, diabetic condition affected liver handling of lipids with higher TAG and HDL secretion into the blood.

## Conclusions

In this paper we presented a series of experiments performed to establish a rat APS as a research tool and demonstrated its application. It can serve as a powerful tool but there are limitations including upper limit of 400 mg/dl for Enite CGM sensor, technical difficulties to set it up initially, and limited number and frequency of experiments due to the duration of each experiment for 2–3 days and concerns of animal welfare. We are currently using a rat APS to study the effects of SGLT2 inhibitors on streptozotocin-induced diabetic rats to assess its benefits and toxicity especially on diabetic ketoacidosis. An APS provides data including total insulin requirement to maintain BGL, average BGL, percent of hyperglycemia, target-in-range, or hypoglycemia during the experimental duration of 2–3 days, from which one can infer the general physiological state of a diabetic rat. A rat APS may be used before, during, or end of drug treatment to assess drug effects. We are also using a rat APS to develop an accurate and adaptable BGL prediction artificial neural network (ANN) model and to test a new ANN-based algorithm for a closed-loop APS as a proof of concept project. We have previously studied a mathematical model of glucose-insulin homeostasis to estimate insulin secretion, glucose uptake by tissues, and hepatic handling of glucose [29] and tracked changes of the parameters of glucose-insulin homeostasis during the course of obesity in mice [30]. A rat APS may provide insights of the progression of both type 1 and type 2 diabetes. In conclusion, a rat APS may be used as a novel tool in addition to a variety of biomedical and engineering approaches to enhance understanding and treatment of diabetes.

## Author Contributions

**Conceptualization:** Guim Kwon.

**Data curation:** Omer Batuhan Kirilmaz, Akshay Radhakrishna Salegaonkar, Justin Shiau, Guney Uzun, Guim Kwon.

**Formal analysis:** Hoo Sang Ko, H. Felix Lee, Guim Kwon.

**Funding acquisition:** Hoo Sang Ko, Guim Kwon.

**Methodology:** Sarah Park.

**Project administration:** Guim Kwon.

**Software:** Sarah Park.

**Writing – original draft:** Guim Kwon.

**Writing – review & editing:** Hoo Sang Ko, H. Felix Lee, Sarah Park, Guim Kwon.

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
