## [Decision Letter · Decision Letter 0]

28 Apr 2021

PONE-D-21-07395

Study of blood glucose and insulin levels in real-time in healthy and diabetic rats using an artificial pancreas system

PLOS ONE

Dear Dr. Kwon,

Thank you for submitting your manuscript to PLOS ONE. After careful consideration, we feel that it has merit but does not fully meet PLOS ONE’s publication criteria as it currently stands. Therefore, we invite you to submit a revised version of the manuscript that addresses the points raised during the review process.

We look forward to receiving your revised manuscript.

Kind regards,

Michael Bader

Academic Editor

PLOS ONE

Journal Requirements:

2) As part of PLOS ONE's publication criteria, the journal requires that in each submission, experiments, statistics, and other analyses are performed to a high technical standard and are described in sufficient detail (https://journals.plos.org/plosone/s/criteria-for-publication). In the case of your paper, we are concerned about the scarcity of details pertaining to your animal research procedures. It is critical to clearly describe all aspects of animal research, animal care and use. To this end, please revise your Methods section to address the following: (a) the number of animals in each group and how you determined the sample size;

(b) the sex and strain of the mice;

(c) all anesthetics and analgesics administered to animals during your study (name of drug, dosage, frequency and route of administration);

(d) details about humane endpoints for any animals who became severely ill during the study;

(e) the rate of mortality during the study and the cause of death (if applicable);

(f) Lastly, please complete and submit the ARRIVE Guidelines checklist (Essential 10 version):

https://arriveguidelines.org/resources/author-checklists.

Reviewers' comments:

Reviewer's Responses to Questions

**Comments to the Author**

1. Is the manuscript technically sound, and do the data support the conclusions?

Reviewer #1: No

Reviewer #2: Partly

2. Has the statistical analysis been performed appropriately and rigorously? 

Reviewer #1: No

Reviewer #2: Yes

3. Have the authors made all data underlying the findings in their manuscript fully available?

Reviewer #1: No

Reviewer #2: No

4. Is the manuscript presented in an intelligible fashion and written in standard English?

Reviewer #1: Yes

Reviewer #2: Yes

5. Review Comments to the Author

Reviewer #1: This manuscript contains several major issues:

1. A number of antidiabetic drugs including insulin formulations have been successfully developed and launched, so far. However, the authors claimed that diurnal variation of blood glucose and insulin levels could disturb evaluation of the effects of antidiabetic agents. The authors should raise detailed case examples.

2. The authors conducted histological assessments and blood biochemical measurements in addition to APS development. The authors should explain about the relationship among results obtained from these experiments. In its current form, the results are not reflected in the title of this manuscript.

3. The authors did not mention whether streptozotocin depleted insulin secretion or not.

4. The authors should conduct statistical analyses of the data shown in Table 1.

The reviewer raised some minor comments below:

5. Please show units in Table 3.

6. In several figures, X-axes are not shown, and Y-axes are fuzzy.

7. The authors should provide a detailed description of intraperitoneal glucose injection (e.g., amount and duration of injection).

Reviewer #2: In the manuscript entitled: “Study of blood glucose and insulin levels in real-time in healthy and diabetic rats using an artificial pancreas system” the authors investigates the use of an artificial pancreas (AP) system on streptozotocin treated rats given two diets with different carbohydrate contents. The AP consists of an Enlite continuous glucose monitor (CGM) (Medtronic), Minimed insulin pump (Medtronic) and an open source controller (Open APS). In addition, the AP and experimental set-up is tested on healthy rats, but without insulin delivery. In addition to the different diets, the rats were injected intraperitoneally with highly concentrated dextrose solutions to simulate large meals. Tissue samples from the pancreas and liver were collected post mortem and analysed for beta-cell content and lipid and glycogen content, respectively.

The manuscripts present interesting information on an animal model suitable for research on glucose metabolism in streptozotocin treated and healthy rats. The authors present several possible areas for which this animal model can be used, and explain that they are themselves using this animal model in their own research. The study includes analysis of tissue samples from both the pancreas and liver, which is a strength of the study. However, the manuscripts lacks a lot of important information in both the description of the methods and results, and it is not suitable for publishing at present. In addition, the manuscript presents the results and subsequent discussions as one text, which at present is not very well structured. I believe that the authors should be able to address both issues adequately.

The authors should include the necessary information in the method and result sections, and I recommend the use of the ARRIVE guidelines for this. They should also make the manuscript easier to navigate for the reader. If they succeed in this, I believe the manuscript would provide useful information for other researcher working in the same field of research.

Comments to the authors:

Major issues:

1. In the title of the manuscript, and several places in the text, you state that you have examined blood glucose and insulin levels in real time. CGM does indeed provide a representation of blood glucose values in (approximately) real-time, but I find the statement of studying insulin levels to be misleading in this context. I believe you have evaluated the insulin dosing calculated by the controller, not the insulin levels present in the rat, which would be impossible in real-time. One could argue that evaluating the insulin delivery from the controller could be an adequate estimation of the actual insulin requirement of the streptozotocin treated rats, but to state that they represent the “physiological truth”, is quite strong. That would imply the perfect controller, which we are all still working on. (But, it would have been great!!) I am sure you have made your thoughts on this and if you could include them in the discussion, it would be nice.

2. What is the benefit of testing the AP on healthy rats? I understand the value of looking at the CGM data and evaluate how an intact pancreas can handle the different diets, and compare this with the controller used on the diabetic rats. However, in the case of the healthy rats, the controller is presented with glucose values from animals with their glucose regulatory system intact. It is then obvious that the OpenAPS suggest continuous low rate infusion of insulin, because the glucose levels do not really change much. With this in mind, how would you argue that you have evaluated the insulin levels of the healthy rats?

3. The results and the subsequent discussions are presented as one text. I am not convinced that this does the manuscript justice, and personally, I prefer them written separately. However, that is not for me to decide but if you choose to keep this way of presenting your findings, an idea could be to divide the section into separate subsections. This will help the reader to find the different results more quickly. Also make sure that the information is written in the appropriate sections, methods in the method sections and results in the result section.

4. The abstract is detailed and describes the major points of the manuscript well. The motivation for the study could perhaps have been more clearly stated. I believe that the aim is to describe the animal model for other researchers to use as a guide when planning their experiments, and not an evaluation of efficacy of the OpenAPS, per se, but a more clearly defined aim would benefit the abstract.

5. In the abstract, you suggest this animal model and technical set-up as a way to study glucose and insulin metabolisms, and different mathematical and engineering models of blood glucose prediction or new algorithms for artificial pancreas systems. However, in your conclusions, you explain that you are using this animal model to study the effect of sodium-glucose cotransporter-2 (SGLT2) inhibitors to treat diabetes type 2. Since this seems to be an important use of the animal model, you should include this kind of use in the abstract as well, as it is a good “selling-point” for the animal model.

6. If I am correct in assuming that you want to describe this animal model for others to use in their research, there is a lot of important information missing both in the method section and in the result section. The crisis of reproducibility is a great challenge to all research, but perhaps particularly in animal research. I strongly recommend you to read and use the ARRIVE guidelines, which is a well-established set of guidelines to help with reporting data from animal research. More and more journals require authors to document that their reporting is in accordance with these guidelines. Below I have outlined some points in your manuscript that need addressing, but if you follow the guidelines you will be able to find more yourself, I’m sure. To start with, it is important to write your planned experiments in the method section, and describe the actual outcome in the result section. The results from this kind of study include much more than just blood glucose values and the amount of insulin used. The results include how the practical aspects of the experiments went, which is often overlooked in publications of animal studies.

a. How many animals did you include in your planned experiments? You should describe this in the method section, both the total amount, the streptozotocin treated and the healthy animals, and how many animals you planned to use for the different experiments. In the result sections, you should start by describing how many rats you ended up using data from. Were any rats excluded? If so, for what reason? It is often a good idea to also include the number of animals in the tables.

b. How was the animal welfare evaluated? Did you use a score sheet? How did you define humane endpoints? This should be described in the method section. In the result section you should describe how the experiments actually went with regards to the animals and their welfare. Did they all tolerate the use of the restrainer, or did they show signs of stress? Did any incidents occur? Any infections?

c. Were some animals used more than once? This is somewhat mentioned in the results section, but not clearly, and should first be stated in the method sections.

d. How long did you plan for each experiments to last? You mention that the rats were not placed in the restrainer for more not more than 3 days, but was that in total or per experiment? Did you have to cut some experiments short? If so, why? Your plans should be stated in the method section, and the actual result should be described in the result section.

e. I do not understand how the rat could be able move freely laterally in the restrainer, so a more detailed description of this would be nice. I have only experience with a restrainer made of more or less solid plastic and not metal bars, as I believe your restrainer consists of after a quick search on the internet. In my experience, the restrainer could not prevent the rats from rotating, but with a small modification of the tube we were able to prevent most of them from turning back-to front. How the rats were fixed during the experiments must be described in more detail in your manuscript, and a photo would be very beneficial.

f. Did you define criteria as to when a rat was considered diabetic? You write in line 107; “Rats developed full-blown diabetes in 1-3 days with BG greater than 500 mg/dl.” (This is actually a result and should be moved to the result section.) The predefined threshold should be stated in the method section and the actual result should be stated in the results section. Were some rats excluded because they didn’t become diabetic, or did they all become diabetic? Did any rats show any side effects of the streptozotocin treatment?

g. When was the different blood samples harvested (HbA1c, TAG, total-cholesterol, and HDL levels)?

h. How was the rats euthanized?

i. The comments and questions above are some points that you need to address, but the ARRIVE guideline will help you report your study and results more clearly. I also recommend you to use the PREPARE guidelines next time you are planning animal experiments. Together these guidelines help us improve our research, and I have great experience in using both of them.

7. What is “controlled conditions” mentioned in line 91?

8. What was the concentration of the isoflurane used for inducing anaesthesia (line 106 and 111)?

9. I am glad to read that you trained the rats so they were familiarized with the restrainer. How and for how long the animals were trained? And how long was the acclimation period before the start of the experiments?

10. When did you calibrate the CGM? Where and how did you collect the blood for calibrations? From the tail as you did for the analysis of TAG, total-cholesterol, and HDL levels? Did you cannulate the tail vein? How you collected blood samples should be described in the method section (and that goes for all blood samples). Did you calibrate the sensors at planned intervals? Was it necessary to calibrate them outside the planned schedule?

11. You state that you use the OpenAPS, but provide no further description. As the OpenAPS is a community based AP system, it constantly changes, as you also mention in the manuscript. You need to state which protocol you used, the date would perhaps be best. You should also state if you did or did not do any alterations or modification of the code/system.

12. In line 81, towards the end of the introduction you write; “An APS will provide a testing platform that is more dynamic and close to natural human physiology”. I believe that you meant to say that the AP systems you describe on live rats resembles human physiology more closely than in silico studies. The UVA/Padova simulator is widely used for in silico studies. It is approved by the FDA as a substitute for animal experiments on DM1 research, and it is based on human data from actual people with diabetes. I would appreciate a discussion of this statement in the manuscript, if and why you would trust the described animal model over, for instance, the UVA/Padova simulator.

13. With which statistical software did you analyse the data? And you should move the description of how you calculated AUC from “Intraperitoneal injection of dextrose” to “Statistical analysis”.

14. In line 204 you write: “These studies suggest that stock concentration of insulin can potentially make a difference in controlling BGL in patients who use APS to treat type 1 diabetes.” Does the OpenAPS take into account which concentration of insulin is used, and make the calculations accordingly? If it does, do you have any ideas why different concentrations would influence the AP ability to regulate glucose?

15. In line 255, you describe how you injected dextrose solution in the IP space. Some of this text is actually a description of the method, and should be written in the appropriate part of the manuscript. You explain that this was performed in order to simulate unannounced meals. Does that mean that you informed the controller every time the rats ate by themselves? This is not described in the manuscript. How did you decide on the amount of carbohydrate in the IP injections? Clearly, they are larger than what rats normally would be able to absorb, as the healthy rats are not able to handle the large glucose load. Would you say that the IP injections of dextrose represents normal conditions (line 281)? Or are they supposed to representations of glucose elevations in humans?

16. What are the units of the values presented in line 270?

17. What are the strengths and limitations of your study? A critical, but fair, self-evaluation of a study should always be a part of any research article.

18. In line 374, you state that these experiments are minimal invasive and can provide general physiological state of rats. This set-up is perhaps not very invasive, but highly interfering. How do you think the fact that the rat is restrained for 2-3 days might affect the results?

19. It is nice to see how the screen of the OpenAPS system looks like and how it is presenting relevant data. However, for this manuscript it does not provide enough data. There is no y-axis for insulin infusions, and the y-axis for BG values are not complete with units. It is nice that you have made a separate figure to present the BG values after IP injections of dextrose. Perhaps the insulin infusions could be included in these figures?

20. It is a strength of your study that you have examined the pancreas and liver post mortem, and present both written results and photos in the manuscript.

21. There is a statement that there are supplementary files, however, these have not been available to me. I apologize if my comments and questions are already addressed in the supplementary files. Have you included supplementary files? And where will you publish the raw data?

Minor issues:

22. The text in the y-axis of some of the figures should perhaps be adjusted, for esthetical reasons

23. This is no big deal, but I believe PLOS One prefers the abstract to be without abbreviations if possible.

24. In the short title, blood is written with a capitol letter but not insulin is not.

25. Is the name of your institution correctly written in line 90? Should “Pharmacy” be written with a capital P?

26. I am not familiar with the term; ”full-blown diabetes” in scientific literature. Is it a commonly used term?

6. PLOS authors have the option to publish the peer review history of their article (what does this mean?). If published, this will include your full peer review and any attached files.

Reviewer #1: No

Reviewer #2: No

---

## [Author Response · Author response to Decision Letter 0]

14 Jun 2021

Editor: Revised manuscript with and without track changes along with new figures are uploaded. ARRIVE Checklist is uploaded as requested. 

Reviewer 1: All suggested comments have been addressed in the revised manuscript. 

Reviewer 2: All suggested comments have been addressed in the revised manuscript.

We sincerely appreciate reviewers helpful comments to enhance the quality of the manuscript.

---

## [Decision Letter · Decision Letter 1]

25 Jun 2021

PONE-D-21-07395R1

Study of blood glucose and insulin infusion rate in real-time in diabetic rats using an artificial pancreas system

PLOS ONE

Dear Dr. Kwon,

Thank you for submitting your manuscript to PLOS ONE. After careful consideration, we feel that it has merit but does not fully meet PLOS ONE’s publication criteria as it currently stands. Therefore, we invite you to submit a revised version of the manuscript that addresses the points still raised by both reviewers.

We look forward to receiving your revised manuscript.

Kind regards,

Michael Bader

Academic Editor

PLOS ONE

Journal Requirements:

Reviewers' comments:

Reviewer's Responses to Questions

**Comments to the Author**

1. If the authors have adequately addressed your comments raised in a previous round of review and you feel that this manuscript is now acceptable for publication, you may indicate that here to bypass the “Comments to the Author” section, enter your conflict of interest statement in the “Confidential to Editor” section, and submit your "Accept" recommendation.

Reviewer #1: (No Response)

Reviewer #2: (No Response)

2. Is the manuscript technically sound, and do the data support the conclusions?

Reviewer #1: Partly

Reviewer #2: Yes

3. Has the statistical analysis been performed appropriately and rigorously? 

Reviewer #1: No

Reviewer #2: Yes

4. Have the authors made all data underlying the findings in their manuscript fully available?

Reviewer #1: No

Reviewer #2: Yes

5. Is the manuscript presented in an intelligible fashion and written in standard English?

Reviewer #1: Yes

Reviewer #2: Yes

6. Review Comments to the Author

Reviewer #1: 1. Several experiments were conducted under non-physiological condition. For instance, in Fig. 6, blood glucose level was over 400 mg/dL, and the authors did not monitored the exact level.

2. The authors might not have conducted statistical analyses properly. In the response to us, the authors mentioned that they used paired t-test and Mann-Whitney U test. Why did the authors analyze with both parametric and non-parametric methods?

3. In Table 1, standard deviation (or standard error) was not shown.

Reviewer #2: The changes made to the manuscript have improved it greatly, still there are a few issues that should be addressed.

1. I appreciate the division of the Results and Discussion section. However, the Result section now begins with; “Establishment of a rat APS system” which is more appropriate in the Method section.

2. Where was the pentobarbital injected?

3. Including figure 1 is very valuable to the manuscript, so thank you for including it. However, I believe is a photo, not a schematic diagram of the rat APS system. In the explanatory text, you write that the rat can move laterally, but have deleted this from the main text based on my previous comment.

4. I am glad to see that you have added the fact that there is concern for the welfare of the animals as a point in your manuscript. In line 225, you write that no experiments were cut short due to “concerns of animal welfare”, and in line 110 and 220 you describe some of the changes/symptoms expressed by the rats. Still, I cannot find your method for evaluating animal welfare or how you defined humane endpoints. If this was not done, I would question if the study was conducted in accordance with legislations and regulations (at least it would not have been in my country). This information should be included in all publications concerning the use of animals, and it is especially vital for this kind of study, which I would argue would be classified as a study with severe impact on the animals.

5. In line 391-392, you make a point of how this animal model provides “a versatile and dynamic in vivo testing platform close to real-life situation”. I agree that this model offers certain advantages for studying different aspects of diabetes and potential treatment options, I would, however, argue that the model does not provide “close to real-life situations”. Of course “close to” can be interpreted in many different ways. However, you are trying to mimic normal human situations. A rat fixed in a restrainer for three days differs greatly from a normal life even for a rat, and definitely from a normal human life situation. Concerning the severe impact on the welfare of the animals included in such a model, I encourage a more realistic presentation of the animal model. However, I appreciate that there would be different views on this topic, and that we might not agree.

7. PLOS authors have the option to publish the peer review history of their article (what does this mean?). If published, this will include your full peer review and any attached files.

Reviewer #1: No

Reviewer #2: No

---

## [Author Response · Author response to Decision Letter 1]

30 Jun 2021

Reviewers 1 and 2: We sincerely appreciate your excellent suggestions and your valuable time in reviewing the manuscript thoroughly.

---

## [Editor Report · Decision Letter 2]

2 Jul 2021

Study of blood glucose and insulin infusion rate in real-time in diabetic rats using an artificial pancreas system

PONE-D-21-07395R2

Dear Dr. Kwon,

We’re pleased to inform you that your manuscript has been judged scientifically suitable for publication and will be formally accepted for publication once it meets all outstanding technical requirements.

Kind regards,

Michael Bader

Academic Editor

PLOS ONE
---

## [Editor Report · Acceptance letter]

6 Jul 2021

PONE-D-21-07395R2 

Study of blood glucose and insulin infusion rate in real-time in diabetic rats using an artificial pancreas system 

Dear Dr. Kwon:

I'm pleased to inform you that your manuscript has been deemed suitable for publication in PLOS ONE. Congratulations! Your manuscript is now with our production department. 

Kind regards, 

on behalf of

Prof. Michael Bader 

Academic Editor

PLOS ONE